# Characteristics of the Two Asian Bumblebee Species *Bombus friseanus* and *Bombus breviceps* (Hymenoptera: Apidae)

**DOI:** 10.3390/insects11030163

**Published:** 2020-03-03

**Authors:** Cheng Liang, Guiling Ding, Jiaxing Huang, Xuewen Zhang, Chunhui Miao, Jiandong An

**Affiliations:** 1Key Laboratory for Insect-Pollinator Biology of the Ministry of Agriculture and Rural Affairs, Institute of Apicultural Research, Chinese Academy of Agricultural Sciences, Beijing 100093, China; liang1087@163.com (C.L.); huangjiaxing@caas.cn (J.H.); 2Institute of Sericulture and Apiculture, Yunnan Academy of Agricultural Sciences, Mengzi 661101, Yunnan, China; zxw216226@163.com (X.Z.); mchanchor@hotmail.com (C.M.)

**Keywords:** *Bombus friseanus*, *Bombus breviceps*, food plants, life history, colony development, mating

## Abstract

This study compared the food plants, life cycle, colony development, and mating behaviour of the two Asian bumblebee species *Bombus friseanus* and *B. breviceps*, which are very important pollinators for many wild flowers and crops in local ecosystems. Both species were shown to be highly polylectic. Differences were observed in their life cycles and colony development patterns. The colony foundation rate of the field-collected queens was high in both species, 95.5% in *B. friseanus* and 86.5% in *B. breviceps*. The intervals from colony initiation to colony sizes of 30, 60, and 80 workers and to the first male and gyne emergence were significantly shorter in *B. friseanus* than in *B. breviceps* (*p* < 0.01). The development period of the first batch of workers showed no significant difference between the two species (*p* > 0.05). Compared with *B. friseanus*, *B. breviceps* produced remarkably higher numbers of workers (135 ± 30 workers/colony in *B. friseanus* and 318 ± 123 workers/colony in *B. breviceps*) and males (199 ± 46 males/colony in *B. friseanus* and 355 ± 166 males/colony in *B. breviceps*) (*p* < 0.01), with notable variation was found among the colonies in both species. With no significant difference in the mating rate between these two species, the copulation duration of *B. breviceps* (1.54 ± 0.63 min) was strikingly shorter than that of *B. friseanus* (27.44 ± 11.16 min) (*p* < 0.001). This study highlights the characteristics of the two Asian bumblebee species and will aid further studies on their conservation and agricultural pollination use.

## 1. Introduction

Bumblebees are distributed mainly in the world’s north-temperate regions [1]. Most bumblebees are annual social insects. Only mated queens survive the winter, hibernating in small cavities. In early spring, they leave their hibernation sites to found new colonies. First, the queen produces a batch of diploid eggs that will develop into the first batch of workers to initiate the colony. These workers assist the queen in raising subsequent broods that first develop into more workers and then into sexual individuals (males and gynes). Then, the mated new queens go into diapause while the mother queen, the males, and the workers die [2].

Bumblebees are important pollinators of many wild plants and crops, so they provide vital pollination services in both natural and agricultural ecosystems [3,4]. The exploitation of the potential of bumblebees for crop pollination should be based on the propagation of natural populations and the domestication. Domestication attempts involve controlling every step of the bumblebee’s life cycle. Commercial rearing started only if the bumblebees had been domesticated [3]. Currently, the main species reared commercially is the *B. terrestris* from Eurasia. To date, the commercial *B. terrestris* has been deliberately introduced into many foreign territories for crop pollination because of its recognized value [3,5]. 

However, the intentional introduction of this alien species has led to biological invasion in many countries [6,7,8,9]. The accidental escapees and intention spread of non-native bumblebees may affect native plant reproduction [10], and these introduced bees may compete for food resources and nesting sites with native species [7,11], spread new diseases and parasites [12,13], and disrupt the reproduction of native bumblebee species [14,15].

Considering that *B. terrestris* would strongly impact the indigenous ecosystems, attempts to rear native bumblebee species have been conducted in many countries, such as *B. impatiens* and *B. occidentalis* in North America [16], *B. ignitus* in South Korea [17,18], *B. hypocrita* and *B. ignitus* in Japan [19,20,21], and *B. ignitus*, *B. lucorum*, *B. pyrosoma*, *B. picipes*, *B. lantschouensis,* and *B. patagiatus* in China [22,23]. Most reared species belong to the subgenus *Bombus* s. str., with few species from the subgenera *Pyrobombus* and *Melanobombus*. 

*B. friseanus* and *B. breviceps*, belonging to the subgenera *Melanobombus* and *Alpigenobombus*, respectively, are distributed mainly in the regions of Southern China, Himalaya, and Southeast Asia [24]. They are also the two most abundant bumblebee species in Yunnan Province of Southwest China [25]. These two Asian bumblebee species are important pollinators for many plants and play key roles in local ecosystems. For instance, *B. friseanus* is an important pollinator for alpine plants, such as *Pedicularis* species [26,27,28] and *Salvia przewalskii* (Lamiaceae) [29,30]. *B. breviceps* is the major pollinator of *Amomum subulatum* [31,32]. In this study, to evaluate whether *B. friseanus* and *B. breviceps* have the potential for domestication and commercial applications, we first surveyed their food plants across Yunnan Province and then monitored their life history in Mengzi city, Yunnan Province. The colony development and the mating behaviour of these two species were also recorded and compared under rearing room conditions.

## 2. Materials and Methods

### 2.1. Investigating Food Plants

From 2002 to 2017, a systematic survey of bumblebees was ongoing in China [1,22]. The survey of bumblebees in Yunnan Province was conducted extensively between 2009 and 2017. Information on the food plants visited by *B. friseanus* and *B. breviceps* was compiled from all the records collected during the survey.

### 2.2. Monitoring the Life History

The life cycles of both *B. friseanus* and *B. breviceps* were observed in Mengzi city, Yunnan Province, between 2016 and 2017. Observations were carried out three times a month, on the 5th, 15th, and 25th, respectively. When it was rainy, the field work was postponed until the next sunny day. For each field observation, we recorded the presence of queens, workers, and males for both species for 1 h.

### 2.3. Bumblebees Laboratory Rearing

During mid-March and early April 2016 and 2017, *B. breviceps* queens and *B. friseanus* queens that had emerged from diapause were collected in the field from Mengzi city, Yunnan Province. Each queen was reared individually in a small “starting box” (wooden box sized 16 × 12 × 16 cm) kept in an identical dark climate room at 29 ± 1 °C and 57% ± 2% relative humidity for nest initiation. The small colony was transferred into a larger “rearing box” (wooden box sized 31 × 24 × 16 cm) when the first batch of workers (usually 4–6 workers) emerged.

During the rearing experiment, the bumblebees were fed ad libitum with sugar syrup (50% sugar content, w/w) and commercially available fresh pollen collected by the honey bee *Apis mellifera*.

### 2.4. Recording the Colony Development

The colony development of *B. breviceps* and *B. friseanus* was tracked by direct daily observation under red light. The developmental traits were recorded, including data on the queen’s first oviposition (colony initiation); the developmental time for the first batch of workers; the times when colonies reached 30, 60, and 80 workers; and the date of the first male and gyne emergence. As soon as the first male or gyne emerged, they were removed from the colony and reared in separate boxes. At the end of the colony’s life, we counted all the workers, males, and gynes produced by each colony. During the observation, we also counted the dead bees in the colonies to determine the total number of workers and sexual individuals produced. The colony lifetime was defined as the date when the colony had approximately 30 workers left after the old queen had died.

### 2.5. Observing the Mating Behaviour

Copulation of both *B. breviceps* and *B. friseanus* was observed in rectangular fine netting cages (2.0 × 1.2 × 1.5 m). The newly emerged queens and males were collected from laboratory-reared colonies and transferred to separate boxes for rearing. For each observation, ten queens at eight days after emergence were released into the mating cages a few minutes after three times as many males (10–15 days old) had been put in. The queens and males present in a mating cage were collected from different colonies to avoid inbreeding. The queens were discarded if they had not mated within 1 h (for *B. breviceps*) or 2 h (*B. friseanus*) after being placed in the mating cage. To investigate the mating duration (the length of time that copulation takes), we recorded the time for 30 mating pairs for both bumblebee species. After completion of the first copulation, we marked mated *B. breviceps* queens and *B. friseanus* queens and put them back in the mating cage to observe if a second mating occurred.

## 3. Results

### 3.1. The Food Plants of B. friseanus and B. breviceps

We recorded a total of 85 plant species belonging to 26 families that were foraged by *B. friseanus* and *B. breviceps* in Yunnan Province. Our survey noted that *B. friseanus* foraged the flowers of 49 plant species across 17 families. There were 47 plant species belonging to 17 different plant families visited by *B. breviceps* (Appendix A). The most common plants foraged by *B. friseanus* included *Trifolium repens*, *Clinopodium megalanthum,* and *Cirsium lidjiangense*, belonging to the families Fabaceae, Lamiaceae, and Asteraceae, respectively. *B. breviceps* was found to forage on *Crotalaria assamica* (Fabaceae) and *Rubus alceaefolius* (Rosaceae) in many places. The major foraging resources of both species were plants of the families Fabaceae and Lamiaceae (Figure 1 and Figure 2). Foraging of *B. friseanus* workers on plentiful species of Scrophulariaceae and Asteraceae was also observed. There were 11 plant species foraged by both *B. friseanus* and *B. breviceps* (Appendix A).

### 3.2. The Life History of B. friseanus and B. breviceps

Comparing the life cycles of the two species, the flying season of *B. breviceps* lasted longer than *B. friseanus*, although the queens of *B. friseanus* emerged from hibernation earlier, and so did the first workers and sexual reproductions. For *B. friseanus*, the hibernated queens started nest searching in early March. Workers were observed to forage during early April until mid-September. The males and gynes appeared during July and September. For *B. breviceps*, the overwintered queens started to found new colonies in late March, and the workers began to forage in early May, which could last until mid-December. The reproductive males were observed during the early September until mid-December and the gynes appeared during the late September until late November (Figure 3).

### 3.3. The Colony Development of B. friseanus and B. breviceps

A total of 22 *B. friseanus* queens and 111 *B. breviceps* queens were collected from the field during the spring in 2016 and 2017. During rearing in the climate room, most queens started to lay eggs and initiate colonies, with no significant difference in the nesting rate between the two species (95.5% in *B. friseanus* and 86.5% in *B. breviceps*; χ^2^ = 0.677, *df* = 1, *p* = 0.411). The duration until first oviposition (the preoviposition period since queens were collected from the field) was 8.9 ± 7.7 days in *B. friseanus* and 13.6 ± 7.0 days in *B. breviceps* (Mann–Whitney *U* test: *p* = 0.052). There was no significant difference in the period until first worker emergence between the two species (Mann–Whitney *U* test: *p* = 0.456). The intervals from colony initiation to colony sizes of 30, 60, and 80 workers were significantly shorter in *B. friseanus* than in *B. breviceps* (Mann–Whitney *U* test: *p* < 0.01). The period up to the first male and gyne emergence was also significantly shorter (*p* < 0.01) in *B. friseanus* (males: 60.6 ± 5.1 days, gynes: 73.5 ± 5.9 days) than in *B. breviceps* (males: 120.4 ± 12.7 days, gynes: 141.2 ± 16.0 days). The colony lifetime of *B. breviceps* was significantly longer than that of *B. friseanus* (217.8 ± 21.7 and 132.5 ± 15.5 days, respectively; *p* < 0.01) (Table 1).

For the first batch of workers, no significant difference was detected in the average developmental duration (26.6 ± 3.2 days in *B. breviceps* and 25.9 ± 1.8 days in *B. friseanus*; Mann–Whitney *U* test: *p* = 0.595). The developmental time of the egg, larvae, and pupae of the first batch of workers also showed no significant difference between *B. friseanus* and *B. breviceps* (*p* > 0.05). Although the egg, larval, and pupal periods of *B. friseanus* (3.4 ± 0.7 days; 11.6 ± 1.6 and 10.9 ± 0.7 days, respectively) were slightly shorter than that of *B. breviceps* (3.6 ± 1.0 days; 11.7 ± 2.6 and 11.3 ± 1.7 days) (Figure 4). 

Even kept under the same laboratory conditions, colonies of *B. friseanus* and *B. breviceps* showed tremendous variation in the number of workers and males produced. In *B. friseanus*, the number of workers per colony ranged from 86 to 202, and the number of males ranged from 109 to 276. Larger variation was detected in *B. breviceps* colonies; 60–663 workers and 108–921 males could be produced per colony (Figure 5). Significant differences in the numbers of workers (135 ± 30 workers/colony in *B. friseanus* and 318 ± 123 workers/colony in *B. breviceps*; *t* = −8.108, *df* = 41.250, *p* < 0.01) and males (199 ± 46 males/colony in *B. friseanus* and 355 ± 166 males/colony in *B. breviceps*; Mann–Whitney *U* test: *p* < 0.01) were observed between these two species. For the number of gynes, no significant differences were detected between the two species (63 ± 21 gynes/colony in *B. friseanus* and 47 ± 31 gynes/colony in *B. breviceps*; Mann–Whitney *U* test: *p* = 0.115) (Figure 5). The field-collected queens of *B. breviceps* produced significantly stronger colonies (721 ± 255 bees/colony) than did those of *B. friseanus* (397 ± 91 bees/colony) (*t* = −6.466, *df* = 45.484, *p* < 0.01). 

### 3.4. The Mating Behaviour of B. friseanus and B. breviceps

There was no significant difference in the mating rate between these two species, with 62.96% in *B. friseanus* and 57.53% in *B. breviceps* (χ^2^ = 1.449, *df* = 1, *p* = 0.229). The copulation duration of *B. friseanus* (27.44 ± 11.16 min) was significantly longer than that of *B. breviceps* (1.54 ± 0.63 min) (Mann–Whitney *U* test: *p* < 0.001) (Table 2). Given the opportunity, re-mating occurred in none of the mated *B. friseanus* queens, while 10.07% of *B. breviceps* queens had a second mating (Table 2).

## 4. Discussion

Both *B. friseanus* and *B. breviceps* were found to forage on plenty of plant species in Yunnan Province, indicating that they are polylectic. These two species were also reported to visit diverse plants in Sichuan Province [24]. *B. friseanus* and *B. breviceps* were distributed in different regions with different flora, reflecting their diverse floral host range for the collection of nectar and pollen. Further detailed studies focusing on the nutrition needs of these species would promote their domestication and commercial rearing system.

In many sites that we surveyed, both *B. friseanus* and *B. breviceps* were the most abundant species, co-existing with many other bumblebee species. Given the species richness at one location, competition or different foraging preferences between species may occur. Foraging bumblebees of one species could avoid flowers already visited by other species in a mixed community. This phenomenon has been reported by Goulson et al. (1998): Both *B. terrestris* and *B. pascuorum* could selectively avoid inflorescences of *Symphytum officinale* already visited by other *Bombus* species [33]. Notably, *B. friseanus* and *B. breviceps* share major foraging resources, and further research is necessary to elucidate the competitive foraging interactions between them. 

Mating duration has been recorded in several bumblebee species. Similar or longer average copulation durations as those of *B. friseanus* have been reported: 42.2 min in *B. bifarius*, 44.7 min in *B. californicus* [34], 30.3–36.9 min in *B. terrestris* [35], 26.3 min in *B. hypnorum* [36], 28.9 min in *B. ignitus*, 27.3 min in *B. patagiatus,* and 20.6 min in *B. lantschouensis* [23]. Considerably shorter durations have been reported for *B. frigidus* (10.2 min) and *B. rufocinctus* (9 min) [34]. Our observed mating duration of *B. breviceps* (1.5 min) is extremely shorter than that of other species reported in previous studies. This is the shortest reported record for mating duration in all bumblebee species. Currently, little is known about the proximate reasons underlying mating duration variations.

So far, most *Bombus* species examined appear to be monandrous [23,37,38,39], with the exception that *B. hypnorum* and a few other species have been confirmed to be polyandrous [40,41]. The occurrence of second mating in *B. breviceps* queens suggests that this species may be another naturally polyandrous species. Without analysis of natural colonies, we are unable to draw solid conclusions based on such a low re-mating rate under laboratory conditions, especially considering that female choice to re-mate may depend upon the previous time spent mating [36]. Previous studies suggest that *B. terrestris* males impose monandry on queens by inserting a mating plug during copulation [35,42,43]; the fatty acids in mating plugs could prevent further mating of *B. terrestris* queens [42]. It was also suggested that the copulation of *B. terrestris* lasted 37 min to match the time required to deposit the ‘mating plug’ fully [35]. Hence, comparison of the mating plug between monandrous species and polyandrous species is needed to shed light on the proximate factors related to mating times and duration.

Nectar robbing refers to the collection of nectar by biting or piercing holes in flowers, often without providing an effective pollination service [44,45]. Flowering plants with long tubular flowers or nectar spurs are most likely to be robbed [46]. Previous studies reported that *Salvia przewalskii* was robbed by *B. friseanus* [29,30]. During our field collection, we also observed that *Salvia leucantha* was robbed by *B. breviceps* (Figure 2). Since nectar robbing can have diverse consequences on the host plants, ranging from detrimental to beneficial effects on plant reproductive success [47], pollination of plants with long corolla tubes should be performed with caution using *B. friseanus* and *B. breviceps*. Additionally, further studies are needed to reveal the effects of nectar robbing on pollinator behaviour and plant reproduction.

Previous studies have already revealed that Chinese *B. lucorum*, *B. patagiatus*, *B. ignitus*, *B. pyrosoma*, *B. picipes*, and *B. lantschouensis* are easy to rear and have promising application prospects [22,23]. The selected *B. patagiatus* and *B. lantschouensis* are being used in trials with fruit crops on local farms in north China and the pollination efficiencies of domestic bumblebees are comparable to that of imported *B. terrestris*. However, there are still some problems, such as low reproductive efficiency, high production cost, and unstable production process, need to be overcome [22]. For now, there is no commercial system developed for the mass production of native bumblebees in China. 

Colony initiation with queens collected in the field is the first step for domestication attempts. Our study showed that the colony foundation rates of *B. friseanus* and *B. breviceps* field-collected queens were 95.5% and 86.5%, respectively. This value resembles that reported in *B. terrestris* [48]. For pollination applications, it is desirable to have strong, long-lasting colonies. Worker productivity is an important criterion in evaluating the potential pollination efficiency of a bumblebee species. A minimum of 50 workers in *B. terrestris* colonies is necessary for providing normal pollination services [3,48]. Striking variations in colony size were detected among colonies in *B. terrestris* [2], *B. lucorum* [49], *B. impatiens* [50], *B. friseanus,* and *B. breviceps* (this study). Colonies of *B. friseanus* and *B. breviceps* could reach an average size of 135 and 318 workers, respectively. The mating success of the progeny queens was also high. These excellent biology traits displayed in different life cycles make these two species promising candidates for domestication and commercial rearing. To facilitate the domestication of *B. friseanus* and *B. breviceps*, further studies on year-round rearing and selective breeding are now in progress.

## 5. Conclusions

In conclusion, we studied the food plants, life cycle, colony development, and mating behaviour of *Bombus friseanus* and *B. breviceps*. Our results showed that these two Asian bumblebee species are highly polylectic. The colony foundation rate of the field-collected queens was high in both species. Although a notable difference was detected in the colony development pattern and mating behaviour between *B. friseanus* and *B. breviceps*, both species could produce colonies with remarkably high number of workers. Our results support that both *B. friseanus* and *B. breviceps* are promising candidates for domestication and commercial rearing. However, further studies on year-round rearing and selective breeding are needed for mass production of these two species’ colonies. 

## Figures and Tables

**Figure 1 insects-11-00163-f001:**
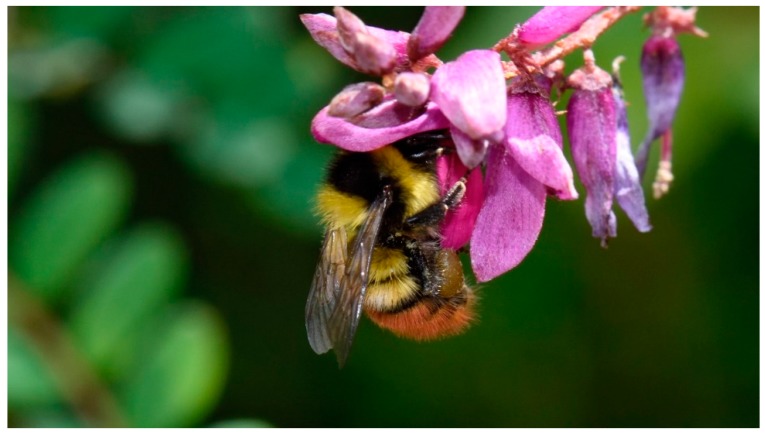
*Bombus friseanus* worker visiting *Indigofera forrestii* (Fabaceae) in Yunnan Province of Southwest China (Photo by Jiaxing Huang).

**Figure 2 insects-11-00163-f002:**
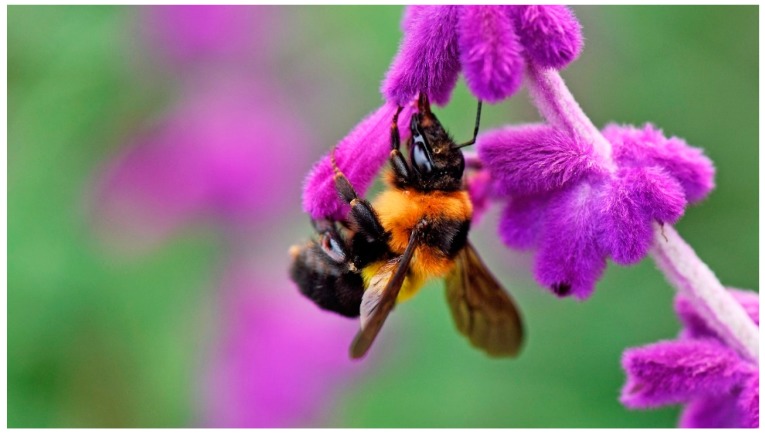
*Bombus breviceps* worker visiting *Salvia leucantha* (Lamiaceae) in Yunnan Province of Southwest China (Photo by Jiaxing Huang).

**Figure 3 insects-11-00163-f003:**
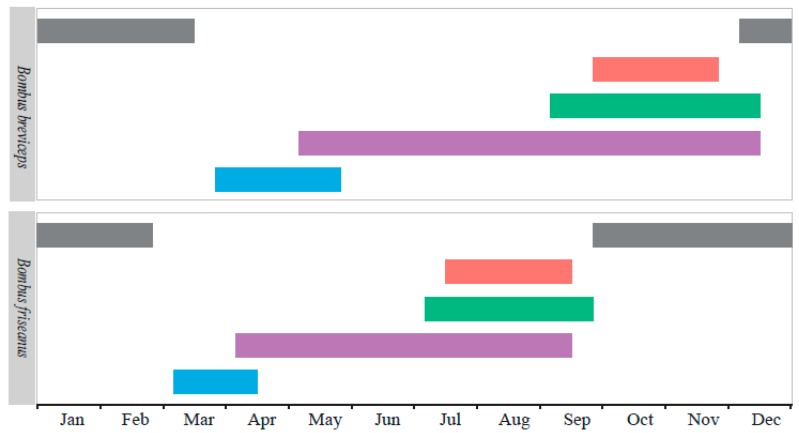
The life history of *Bombus friseanus* and *Bombus breviceps* in the field in Mengzi city, Yunnan Province, China. Blue bars (

) indicate the seasonal occurrence of hibernated queens, purple bars (

) for workers, green bars (

) for males, red bars (
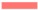
) for gynes, and gray bars (
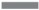
) for mated queens in hibernation.

**Figure 4 insects-11-00163-f004:**
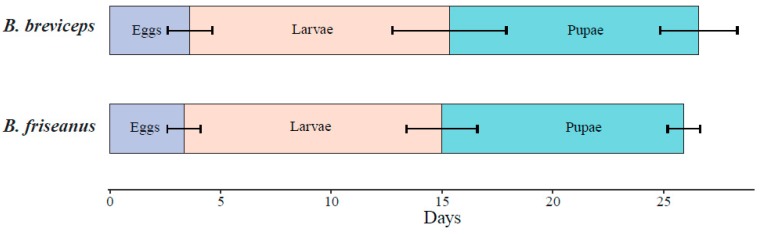
Developmental time (days) of the first batch of workers in *Bombus friseanus* (N = 14) and *Bombus breviceps* (N = 45). Horizontal bars indicate the standard deviation.

**Figure 5 insects-11-00163-f005:**
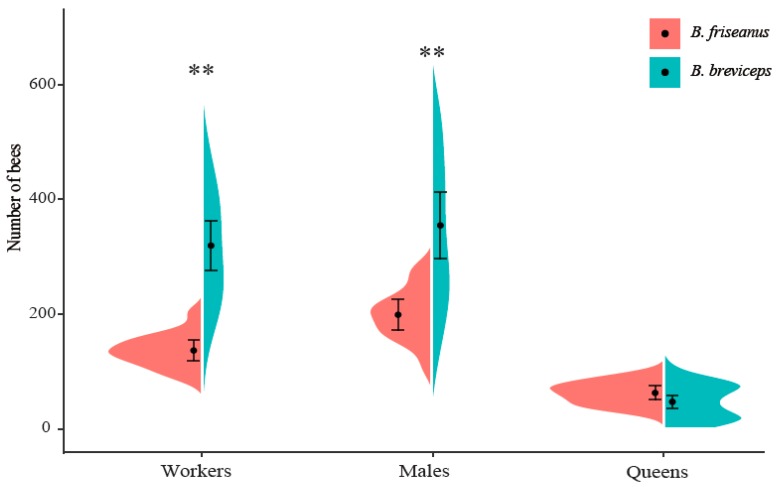
Colony size of *Bombus friseanus* (N = 14) and *Bombus breviceps* (N = 34) under laboratory rearing conditions. Black dots indicate the mean values and horizontal bars indicate the standard deviation. Asterisks represent level of significance. ** *p* < 0.01.

**Table 1 insects-11-00163-t001:** Comparison of duration (days) up to different phases between *B. friseanus* and *B. breviceps* colonies initiated by field-collected queens reared in the laboratory.

Bumblebee Species	Number of Colonies	Initial Oviposition	First Worker Emergence	Colony Size of 30 Workers	Colony Size of 60 Workers	Colony Size of 80 Workers	First Male Emergence	First Gyne Emergence	Colony Lifetime
*Bombus friseanus*	13	8.9 ± 7.7 a	25.4 ± 1.8 a	38.8 ± 5.0 a	46.2 ± 5.2 a	53.6 ± 5.5 a	60.6 ± 5.1 a	73.5 ± 5.9 a	132.5 ± 15.5 a
*Bombus breviceps*	20	13.6 ± 7.0 a	26.5 ± 8.1 a	62.1 ± 13.2 b	76.2 ± 15.0 b	87.8 ± 17.8 b	120.4 ± 12.7 b	141.2 ± 16.0 b	217.8 ± 21.7 b

Same letters in the same column indicate no difference at *p* > 0.05 level and different letters indicate significant difference at *p* < 0.01 according to the Mann–Whitney *U* test.

**Table 2 insects-11-00163-t002:** Mating behaviour of *B. friseanus* and *B. breviceps*.

Bumblebee Species	Mating Success		Mating Duration		Queens Re-mated	
	Number of Queens	Mating Rate (%)	Number of Mating Pairs	Mating Duration ± S. E. (min)	Number of Queens Observed	Queens Re-mated (%)
*Bombus friseanus*	259	62.96 a	30	27.44 ± 10.97 a	149	0
*Bombus breviceps*	216	57.53 a	30	1.54 ± 0.62 b	136	10.07

Same letters in the same column indicate no difference at *p* > 0.05 level and different letters indicate significant difference at *p* < 0.01.

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
