# Peer review of "Characteristics of the Two Asian Bumblebee Species Bombus friseanus and Bombus breviceps (Hymenoptera: Apidae)"

_insects, 2020, doi:10.3390/insects11030163_

Round 1
Reviewer 1 Report
Recommendation Summary
In this paper, the compared the food plants, life cycle, colony development and mating behaviour of two Asian bumblebee species Bombus friseanus and B. breviceps. Both species are relevant for the local ecosystem pollination of wild plants and crops. Based on the background presented, this study is important to determine if both Bombus species can be successfully used for commercial applications. The experimental design was appropriated for the dataset and for the four aspects of the two species highlighted by the authors. However, there are minor issues with the connection between the aspects of the species (food plants, life cycle, colony development and mating behaviour) and their importance or appropriateness for the commercial use and what this commercial use would specifically be. Therefore, minor revision of the manuscript is required to clarify this connection on the introduction and discussion, before this study is published.
Minor issues:
Page 2, line 64:
What would good traits be? The focus in the introduction is on the ecological traits of bumblebees, and then we have the appearance of good traits for commercial use. How are bumblebees commercially used in china? Is this a common practice? How important is this for the local community? Give one example of potential commercial application, like: crop pollination, which is one of the main source of income in the region, if that is the case. Draw a parallel between your observations and the traits for commercial use. That’s why would be clearer if you highlighted in the introduction, some specific commercial use and which bumblebee traits would be best for it.
Page 3 line 114:
Was there any overrepresentation of a food plant in either species and if yes, is that plant visited by both? This would be interesting, since it could indicate preferences, which could lead to other behavioural studies, such as chemical attraction, pollen quality/preference, etc.
Page5 Figure 3:
Add a legend with colours and description of what they mean. It is difficult for the reader to read the figure and identify the meaning of everything in the figure caption.
Page 5. Line 145:
Did you think in drawing a correlation between food plant availability and preferences, and also between duration of hibernation and the intervals from colony initiation to colony sizes and the colony lifetime? This could be interesting to better understand the biological differences between both species by identifying a potential causal relationship. It seems to me that something is causing the/or associated to lower interval of colony initiation and colony size, the number of queens sampled (which may indicate low abundance) and colony lifetime of B. friseanus in comparison to B. breviceps. In terms of commercial use this may indicate that only one species is suitable.
Page 5, Line 162
Delete the following sentence as it is repeated: 'For the first batch of workers, no significant difference was detected in the average'.
Reviewer 2 Report
16: shown to be highly polylectic, foraging on diverse plants - tautological
35: they do not leave hibernation sites often in spring but always
Fig. 2: Obviously this specimen is nectar robbing
136: You wirte: "For B. breviceps, the overwintered queens founded new colonies in late March, and the workers began to forage in early May, which could last until mid-December. The reproductive males and gynes were observed in early and late September, respectively (Figure 3). But in figure 3 you show B. breviceps males and females to live till November resp. December. What is the data base for this figure?
162: ... was detected in the averageFor the first...
You should discuss, if the nectar robbing behaviour, that occurs in B. friseanus and Bombus breviceps (own observations) is a factor that might make these species inapproriate for pollination of some long-tubed flowers. Unfortunately your flower visitation data do not include the activity of the specimens at the flower (nectar collecting, pollen collecting, nectar and pollen collecting, nectar robbing). It should be at least discussed and it should also be the topic of further investigations.
You write in [60], that B. friseanus is an important pollinator for alpine plants: Are there cultivated plants needing bumblebee pollination inj these altitudes? It might be difficult to transfer species in other elevations.
